# The Understanding and Translation of De 德 in the English Translation of the *Daodejing* 道德經

**Xiaojiao Cui**

School of Philosophy, Beijing Normal University, Beijing 100875, China; xj.cui@bnu.edu.cn

**Abstract:** This article investigates the translation of De 德 in the English translation of the *Daodejing*, compares and analyzes several representative translations, and tries to present the complexity and richness of the meaning of De in the thought of the *Daodejing*. The article is divided into three parts. First, it briefly traces the concept of De back to the Shang 商 and Zhou 周 periods, thus laying the foundation for subsequent study. Second, taking Chapter 51 of the *Daodejing* as an example, it explores the meaning of "virtue", which is the most important and common translation of De, in the context of the *Daodejing* and examines related terms such as "potency" and "inner power". Finally, two representative translations of "Xuan De" 玄德 are examined and discussed.

**Keywords:** De 德; Xuan De 玄德; *Daodejing*



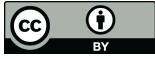

## 1. Introduction

In the world of early Chinese thought, De 德 is undoubtedly an extremely important and difficult concept to understand. The importance and complexity of De is of course due, on the one hand, to its own rich philosophical implications and interpretative possibilities and, on the other hand, to the fact that its connotations and interpretations have often differed among different schools of thought or philosophers, containing a wealth of "internal differences" ([Zheng 2009](#), p. 7). Therefore, how to interpret and translate De in early Chinese philosophical classics has always been a thorny issue. Due to the diverse meaning of De, many translators choose to keep the Chinese phonetic transcription (拼音 *pin yin*) and leave it untranslated. Such an approach, of course, would suggest to readers that De is a complex concept that is extremely difficult to understand and to translate accurately. Alternatively, some might interpret it instead as, when trying to translate it, we are bound to miss the whole picture and even form some kind of misunderstanding. However, from another perspective, as we know, translation itself is an activity of understanding and interpretation, clarifying and determining the meaning of its objects.[1] Therefore, perhaps the suspicion and suspension of translation indicates that the translator has given up the in-depth investigation and precise grasp of the concept to a certain extent. Thus, many translators try to deliver a philosophical understanding and translation of De to reveal its theoretical significance.

In the *Oxford English Dictionary*, De is explained under the category of Daoism and Confucianism as follows: "In Taoism, the essence of Tao inherent in all beings", and "In Confucianism and the extended use, moral virtue". Similarly, in the studies and translations of Chinese classics published thus far, De in Confucian texts is usually translated as virtue to emphasize its moral connotation.[2] In contrast, in Daoist texts, the interpretation and translation of De are more complicated and diverse, with common translations such as character, power, inner power, potency, virtue, efficacy, and integrity.[3] The *Daodejing* tends to pair De with Dao 道 and expound De in the context of the relationship between Dao and myriad things (*wan wu* 萬物). Consequently, to accurately grasp De in the *Daodejing*, it is necessary to place De in the theoretical context of Dao, myriad things (*wan wu* 萬物), non-action (*wu wei* 無為), and self-so (*zi ran* 自然), and examine the significance of

De therein. In fact, the various translations of De in the English versions of the *Daodejing* show such an effort of understanding and interpretation.

This article will first trace the origins of the character De, briefly reviewing and discussing the transformation and evolution of the meaning of De in the Shang 商 and Zhou 周 periods. It will then investigate some representative English translations of De in the relevant chapters of the *Daodejing* and explore the rich meanings of De through these translations. Finally, it will examine the translations of Xuan De, which is without any doubt one of the most important concepts that is related to De in early Daoism.

## 2. De in the Shang and Zhou Periods

The description of De in the *Daodejing*, although unique, is not created out of nowhere but has its origins. As early as the time of the *Shang* and *Zhou*, De was already an important concept in the Chinese intellectual and political world, and some scholars have thus summarized the period of Shang and Zhou as the "Age of De" (Zheng 2009, p. 21). Along with the drastic changes in the ideological and sociopolitical systems at the time of the Shang and Zhou, the connotation of De had also evolved and had become more complex and richer in the long-term cultural accumulation.

The first occurrence of De is very early, but scholars have not formed a unified view as to whether the original graph already existed in the oracle bone graphs and whether the concept of De was initially formed. However, it is worth mentioning that, according to the textual research and references of previous scholars, the Shang oracle bone graph 𔓏 (jiaguwen heji 甲骨文合集 7268) and its variants are related to the character De in the *Zhou* bronze inscriptions and are most likely the initial graph for De[4] (Jao 1978, pp. 77–100; Nivison 1978–1979; Xu 1989, pp. 168–69; Guo 2019; Ye 2022, pp. 51–56).

For the meaning of the oracle bone graph 𔓏, the common interpretations are virtue (*de* 德)[5], get (*de* 得) (Jao 2009, p. 233), patrol (*xun* 巡, *xing* 省)[6], and expedition (*zheng* 征) (Nivison 1978–1979). According to Zheng Kai 鄭開, it is quite possible that the original meaning of De is related to the regular inspection of the Four Directions (*si fang* 四方) by the Son of Heaven (*tian zi* 天子) in the Shang and Zhou periods, an activity known as "*xing fang* 省方" in the Shang Dynasty and "*yu xing* 遹省" in the Zhou, since De is sometimes transcribed as 眚 and taken as a phonetic loan for *xing* 省. Importantly, *xing fang* and *yu xing* are not merely equivalent to forceful conquest; rather, they include patrols, hunting and expeditions and manifest the virtue of the Son of Heaven. In other words, the purpose of *xing* 省 is not to demonstrate the power of the ruler and threaten the Four Directions. Rather, it is to display the ruler's virtue and thus to edify the Four Directions. To this extent, *xing* accommodates a more tender dimension and is permeated with political rationality (Zheng 2009, p. 140). It is precisely this humanistic kernel of *xing* that nourishes a kind of ethos (Chen 1996, pp. 296–99) and gradually nurtures the notion of De, which includes ideological content based on a high degree of humanistic rationality and an abundance of ethical principles.

In the bronze inscriptions of the Western *Zhou* period, the occurrence of De and its variants is very common (Liu 1986; Ye 2022, pp. 66–68). In the *Book of Changes (Zhou Yi* 周易), the *Book of Odes (Shi Jing* 詩經), the *Book of Documents* (*Shang shu* 尚書), and other early literary sources, not only is De frequently discussed, but some related customary phrases (*guan yu* 慣語), such as bright virtue (*ming de* 明德), honoring virtue (*jing de* 敬德), and corrective virtue (*zheng de* 正德), are also found. Thus, it can be inferred that De had already become a very popular ideological term at that time. Compared with De in the Shang dynasty, which was an era when the term was understood and interpreted more at the level of bureaucracy, with the rise of humanistic rationality at the time of the Western Zhou, De also went through a process of shifting from external political systems or behaviors to internal virtues. An obvious example is Zhou's reflection and interpretation of the major historical event, King Wu conquered Shang (*wu wang ke shang* 武王克商), which encompasses great significance in the history of the Zhou dynasty and even in the political

history of China. Being a small state, the Zhou was vastly outmatched by the Shang, but it eventually won the battle against the powerful Shang. The Zhou people believed that the legitimacy of the Zhou's victory over the Shang and its mandate from heaven came from the fact that "heaven's mandate is not always constant" (*tian ming mi chang* 天命靡常) (Cheng 2004, pp. 406–8) and that Heaven (*tian* 天) has no bias but facilitates those who are virtuous (*huang tian wu qin, wei de shi fu* 皇天無親，惟德是輔) (Kong 2004, pp. 659–64).[7] In the worldview of the Zhou people, De and the Mandate of Heaven form a two-way internal and external interaction, enabling people to gain the favor of Heaven through their own virtuous cultivation and their esteem of De. De, on the one hand, symbolizes the human inheritance of the Mandate of Heaven[8] and, on the other hand, signifies that this inheritance in fact originates from humanity's intrinsic virtue and that there exists a sort of "religious-cum-ethical connection" between virtue and the Mandate of Heaven (Hou 1980, p. 26).

In addition, as Ye Shuxun 葉樹勳 has keenly observed, compared with that on the Shang oracle bones, the graph for De in Western Zhou inscriptions has added the "*xin* 心" (heart/mind) element and is transcribed as 悳. Such a visual change is not accidental but is meant to emphasize the meaning of inner virtue that De implies (Ye 2022, p. 73). Roger Ames has also pointed out that, since 悳 encompasses both the elements of *zhi* 直 and *xin* 心, it may also be best understood from the meaning of the *zhi* element, and the fundamental meaning of *zhi* is to grow straight without deviation in the context of organic occurrence. The organic dimension of *zhi* is underscored by its cognates, *zhi* 植, to sow and *zhi* 植, to plant (Callicott and Ames 1989, p. 125). According to Ames, it is very reasonable to infer that De in the form of 悳 might imply the growth of one's inner virtue. Interestingly, Constance Cook has observed that, in Zhou bronze inscriptions, De sometimes also has a *bei* 貝 element, which represents a cowry shell. In the Zhou period, cowry shells had a particular role as sacred objects in gift-giving and were closely associated with the virtue (*de* 德) of those who gave them or used them in sacrifice (Cook 1997).

The internalization and spiritualization of De at the time of the Yin and Zhou dynasties are, without any doubt, accompanied by the philosophizing process of De as an ideological concept and the further refinement of its meaning. By the time of the rise of the Hundred Schools (*bai jia* 百家), De no longer merely denoted moral virtues, nor did it just pervade through various virtues (e.g., filial piety, parenthood, beneficence, respect, and tenderness); it was increasingly converging on benevolence (ren 仁) and righteousness (yi 義) under the dominance of Confucianism while, at the same time, retaining and developing the complex meanings of nature, character, and spiritual experience in the traditions of the Taoists and the Yin–Yangists (Zheng 2009, pp. 13–14).

In short, in the Shang and Zhou, along with the changes in politics and thought, De accumulated different levels of meaning over a long process of conceptual evolution and consolidated an important foundation for the occurrence and development of the philosophical concept of virtue in the period of the Warring States (*zhan guo* 戰國). In addition, the denotation of De in the *Daodejing* is undoubtedly complex, and the complexity is of course due to the multiple meanings that it carries and the various interpretations that it contains; however, this complexity is also due to the profound theoretical relationship between De, Dao, and thinghood, and that between De, heart/mind, and nature (*xing* 性). The complexity of De in the *Daodejing* also manifests in the differences in meaning and understanding between the different terms used in translating De in the various translations of the *Daodejing*.

### 3. De as Virtue in the Translation of the *Daodejing*: An Example from Chapter 51

*3.1. De as Virtue*

In the translation of Confucian texts such as the *Analects* and *Mencius*, the translation of De as virtue is extremely common and hardly debatable. Indeed, in Confucian texts, the meaning of De is relatively clear. Although Confucianism distinguishes between different morals, such as benevolence and righteousness, in general, De refers to various moral norms and their presence in one's inner consciousness. In this sense, the translation

of De as virtue is also very straightforward and will hardly cause misunderstanding among readers in the English-speaking world. Obviously, the word De in Confucian texts signifies virtue and virtuous behavior; as the *Collins Dictionary* explains, "A virtue is a good quality or way of behaving"[9].

Interestingly, when examining the translations of the *Daodejing*, the translation of De as virtue is in fact also quite common. In the early days of classical translation, it was not uncommon to see matching meanings (*ge yi* 格義) based on the need for understanding and dissemination, with Dao and De being translated as God and virtue, respectively. In addition to matching meanings, the lack of or deviation from understanding caused by the initial exposure to a heterogeneous textual tradition is also one of the major reasons for the translation of De as mere virtue. For example, the translation published by D.T. Suzuki and Paul Carus in 1913 translates the title of the *Daodejing* as "The Canon of Reason and Virtue" (Suzuki and Carus 1913), a title that was mentioned earlier in the writings of the American missionary Samuel W. Williams (Williams 1848). Examining the relevant parts of these two translations, it is easy to see that, in the translators' understanding, the meaning of De in the *Daodejing* is virtue or morality.

After this stage, both the earlier James Legge (Legge 1891) and the later Arthur Waley (Waley 1958), D.C. Lau (Lau 1963), and Wing-tsit Chan (Chan 1969), as well as the more recent Roger T. Ames (Ames and Hall 2003) and Hans-Georg Moeller (Moeller 2007), translate the notion of De as virtue in some chapters of the *Daodejing* as well. It must be noted, however, that with the advancement of Sinological studies, these scholars have a deeper and richer understanding of the *Daodejing* and the philosophical notions therein than nineteenth-century translators such as D.T. Suzuki.[10] Therefore, even though these translators also translated De as virtue in some chapters, in their translations and understandings, the meaning of virtue is not as uniform as that in the translations of Confucian texts. It does mean morality in some chapters[11], but in many chapters, it is closer to the Latin root *virtus* or the Ancient Greek Aretē[12], which refers to nature (*xing* 性), essence, potentiality, potency, etc. The article will focus on this sense of virtue first.

Examining the early classical texts, such as the *Analects*, *Daodejing*, and *Zhuangzi*, *xing* 性 (nature, essence, and concepts alike) are not yet philosophical concepts that are explicitly thematized. The *Analects* says, "The Master's cultural brilliance is something that is readily heard about, whereas one does not get to hear the Master expounding upon the subjects of *xing* 'human nature' or *tiandao* 天道 'the Way of Heaven'" (*Analects* 5.13).[13] The *Daodejing* and the inner chapters of the *Zhuangzi* never mentioned the notion of *xing* either. However, this does not necessarily mean that these texts lack insight and reflection on *xing* or the nature of things, especially the *Daodejing*. It can be said that *Daodejing*'s reflection on *xing* or nature is precisely embodied in De, and scholars of the previous generation, such as Gao Heng 高亨, Xu Fuguan 徐復觀, and Zhang Dainian 張岱年, have pointed out the theoretical correlation between De and *xing*, arguing that De refers to the nature or attributes of all things (Gao 2011, p. 27; Xu 2005, p. 253; Zhang 1982, pp. 23–24) or "the inner foundation for the growth of all things" (Zhang 1989, p. 154).

### 3.2. An Example: De in Chapter 51 of the Daodejing

The meaning of De as *xing* or nature is fully embodied in Chapter 51 of the *Daodejing*, which says:

> The Dao generates them, and the De nourishes them; Material embodies them, and the propensity of every circumstance completes them.

> For this reason, among the myriad things there is none that does not honor the Dao and appreciated the De.

> The honor paid to the Dao and the appreciation owed to the De is not like the bestowal of honors but is ever according to what is so of itself.

> So is that the Dao in generating them, and the De in rearing them, and what grow them, nurture them, shelter them, protect them, nourish them, cover them.[14]

道生之，德畜之，物形之，勢成之。

是以萬物莫不尊道而貴德。

道之尊，德之貴，夫莫之命常自然。

故道生之，德畜之；長之育之；亭之毒之；養之覆之。

This chapter is key in understanding the philosophical meaning of De in the *Daodejing*. However, due to the complexity of its meaning, some translators simply place De in this chapter under the category of Dao instead of translating it and collectively refer to the two as "the Tao", for example:

The Tao gives birth to all beings,

nourishes them, maintains them,

cares for them, comforts them, protects them, takes them back to itself,

creating without possessing,

acting without expecting,

guiding without interfering.

That is why love of the Tao

is in the very nature of things. (Mitchell 1995, p. 79)

In fact, such an understanding and translation have their origins in the history of the interpretation of the *Daodejing*. Cheng Xuanying 成玄英, in his Tang 唐 period commentary on the *Daodejing*, writes the first line in Chapter 51, "the Dao generates them, and the De nourishes them" (*dao sheng zhi, de xu zhi* 道生之，德蓄之), as "the Dao generates and nourishes them" (*dao sheng zhi xu zhi* 道生之蓄之), and explains that "the reason I mentioned only the Dao without saying anything about the De is because the De does not deviate from the Dao; therefore, I omitted the De. As the *Xishengjing* 西升經 said, 'the Dao and the De are not separate, they are mysterious and similar'." Although Cheng's understanding or interpretation suggests the theoretical affinity between Dao and De, it completely erases the importance of De as an independent concept in the *Daodejing* and even in Chinese intellectual history; the "systematic thinking" (Wagner 2003, pp. 78–80; Zheng 2016) that may be implied by placing Dao and De in parallel expressions in the *Daodejing* has also been overlooked. It can even be said that the reader can hardly find De in such an interpretation the text.

Nevertheless, most scholars and translators still consider De in Chapter 51 as a separate philosophical notion from Dao and try to translate and gloss it. For example, D.C. Lau, Wing-tsit Chan, and Philip J. Ivanhoe translate De here as virtue, but they do not clarify the meaning of virtue in their translations[15] (Lau 1963, p. 58; Chan 1969, p. 163; Ivanhoe and Norden 2000, p. 183). Other translators try to juxtapose De and Dao and examine and translate virtue in the context of the relationship between Dao and things, following the ancient explanation that De 德 means what is obtained (*de* 得)" (德者，得也). Thus, these translators identify De as the universal property or individual nature of all things obtained from Dao and translate De as "the power of the Dao" (Arthur Waley 1958, p. 21), "the latent power" (Schwartz 1985, p. 202), "efficacy" (Ames and Hall 2003, p. 156; Moeller 2007, pp. 120–21), "the life force" (Ryden 2008, p. 107), and "the potency" (Graham 1989, p. 218). In general, these translations can be regarded as variations of De in the sense of nature and potency, each with its own emphasis.

First, these translations come specifically from the translator's overall grasp of the *Daodejing*, reflecting a deep understanding of the theoretical connection between De, Dao and thinghood. Second, they have their basis and origin in Chinese intellectual history. Reading De as "what is obtained from Dao" is often seen in many Daoist texts; for example, Chapter 12 of the *Zhuangzi* contains the following passage: "Things got hold of it and it came to life, and it was called De" (Watson 2013, p. 88)[16]. Similarly, Chapter 9 of the *Heguanzi* 鶡冠子 contains the passage, "The way of the Sage is mutually obtained and completed with *Shen Ming* 神明, so it is called Dao and De" (Huang 2014, p. 94). In his

commentary on Chapter 51 of the *Daodejing*, Wang Bi 王弼 writes, "Dao is the origin of things. De is what things obtained (from Dao)"[17]. It is evident that De denotes, on the one hand, the universal nature that all things obtained from Dao and, on the other hand, the essence of the existence of all things.

More importantly, these translations are also influenced by ancient Greek philosophy. In the *Metaphysics* Z, Aristotle takes up the promised study of substance. In Book Θ, he introduces the distinction between matter and form synchronically, applying it to an individual substance at a particular time. When describing the matter and form of substances, he uses the categories potentiality (*dunamis*) and actuality (*entelecheia*) or activity (*energeia*). The term *dunamis* refers to "the origin of movement and change in substance" (Nie 2017). Aristotle distinguishes two different senses of *dunamis*. In the strictest sense, a *dunamis* is the power that a thing has to produce a change. A thing has a *dunamis* in this sense when it has within it "a starting point of change in another thing or in itself insofar as it is other". The exercise of such a power is a *kinêsis*—a movement or process. (Θ.1, 1046a12; cf. Δ.12). There is also a second sense of *dunamis* which might be better translated as potentiality. According to Aristotle, a *dunamis* in this sense is not a thing's power to produce a change but rather its capacity to be in a different and more completed state. (Anagnostopoulos 2011, pp. 388–452). According to the explanations of Chen Kang 陳康 and Nie Minli 聶敏理, Aristotle's use of *dunamis* to describe substance in fact includes a threefold meaning: "capacity, possibility, potentiality" (Chen 2017), or "capacity, possibility, multiplication" (Nie 2017). Here, capacity signifies the ability of something to cause movement or change, as well as the ability to receive movement or change. When this ability is not yet utilized and is in a potential state, it is called potential. It is not difficult to discern similar traces of this idea in the Western academic interpretation of De in the *Daodejing*. For example, A.C. Graham translates De as potency, on which Sarah Allan comments:

> A.C. Graham, who translated de as "potency" in the *Zhuangzi*, noted that it means virtue in the sense that "the virtue of cyanide is to poison," that is, like the Latin root *virtus*, it refers to something intrinsic. (Allan 1997, p. 101)

It is in the sense of nature and potential that Graham and Allan interpret De—cyanide has a very poisonous nature, and, therefore, has the possibility or potentiality of becoming a poison. Similarly, whether De is translated as virtue, the power of Dao, the latent power, efficacy, the life force, or potency, the underlying ideas are identical, and their intention is to show that De, as the nature or potential given to all things by Dao, contains the ability to nurture and develop all things. Moreover, there is no doubt that the connotation of De as potentiality or possibility is indeed reflected in the formulation of De in the *Daodejing* (Ye 2013).

Finally, Chad Hansen's understanding and translation of De is noteworthy as well. Based on the relationship between Dao and De, Hansen translates De as virtuosity. According to his explanation, everything has physical structures that can relate to the internal and external worlds, and it is this physical structure that constitutes De. At the same time, De guides all things to follow the Dao and unfold themselves in their interaction with the world and can appropriately reflect the operation of the Dao in all things and in the world. From this perspective, De is the embodiment of the virtuosity of the Dao (Hansen 2020, p. 282). Chad Hansen's understanding and translation focuses on showing the relationship between the Dao and De and describing how De manifests the Dao. However, it must be mentioned that the individual dimension or personalized principle of De embodied in all things should not be neglected. The notion of self-so (*zi ran* 自然) and the construction of Daoist political philosophy and ethics are based on the significance and role of De in the existence and development of individuals. As Yang Guorong 楊國榮 mentioned in his discussion of the relationship between the Dao and De, De demonstrates more of the principle of individuality, which is manifested both in all things and in the *Daodejing*'s understanding of the relationship between the Dao and human beings (Yang 2021, p. 20). The *Daodejing* affirms the process by which all things obtain the Dao from De but also demands that all things constantly return to the Dao, which is indeed an expression

of the intention to bridge the principle of unity and the principle of individuality. Only by fully recognizing the affirmation of De for the individuality of all things can we truly connect the ontology of Daoist philosophy with political philosophy and ethics.

### 4. *Xuan De* 玄德: Dark Efficacy or Dark Virtue?

In addition to appearing as an independent concept in the form of a single word in the *Daodejing*, De is also often combined with other single words to form philosophical terms, such as *Xuan De* 玄德, *Chang De* 常德, *Guang De* 廣德, *Jian De* 建德, and the most important of which is *Xuan De* 玄德. Just as De as a philosophical concept has its origins, the proposal of *Xuan De* also has a far-reaching and precise background in the history of thought. This background can be regarded as a microcosm of the transformation of the old moral and ritual (*de li* 德-禮) ideology of the Western Zhou period to the new Dao-Law (*Dao-fa* 道法) ideology of the Warring States Period and an innovation on top of the old institutional structure dominated by ethics, morality, rites and music. As Zheng Kai 鄭開 has keenly observed, "the *Daodejing*'s proposal of 'Xuan De' is intended to show that it is a more far-reaching, fundamental, and philosophical 'virtue' than the 'Ming De' 明德 (bright virtue) of the Western Zhou Dynasty, which is obviously a creative interpretation and transformation" ([Zheng 2019](), pp. 13–14). Regarding Xuan De, the *Daodejing* Chapter 10 contains the following passage:

> To give birth to it, to rear it,
>
> to give birth to it without possessing it,
>
> to let it grow without commanding it,
>
> this is called: *Xuan De*. ([Moeller 2007](), p. 25. Slightly modified)
>
> 生而不有，長而不恃，為而不宰，是謂玄德。

The philosophical thought in the *Daodejing* is characterized by a tendency to think about the affairs of the human world through the movement of nature and the Way of Heaven (*tian dao* 天道), and this is also revealing in the *Daodejing*'s understanding of Xuan De. In the *Daodejing*, Xuan De is first used to describe how the Dao treats all things. The Dao ensures the realization of the self-so (*zi ran* 自然) of all things through self-restraint ([Wang 2010]()) and nonaction (*wu wei* 無為). These principles extend to political governance, as *Daodejing* Chapter 81 describes, "The way of Heaven is to benefit and not to harm. The way of the sages is to do good but not to strive for it" (*Tian zhi dao, wei er bu zheng. Sheng ren zhi dao, li er bu hai.* 天之道，為而不爭；聖人之道，利而不害。), in which the sage kings or rulers ensure the realization of the self-so of the people by emulating the nonaction of the Dao.

The English translation of *Xuan De* can be roughly divided into two types of representative ideas in translation and interpretation. First, the De in Xuan De is translated as efficacy, while Xuan De is translated as profound efficacy or dark efficacy, for example, in the translations by Roger Ames, Hans G. Moeller, and Edmund Ryden. Secondly, the De in Xuan De is translated as virtue, while Xuan De is translated as dark virtue, for example, in the translation by D.C. Lau.

Roger Ames translates Xuan De as profound efficacy and explains that the world is created in the collaboration of focus and field, event and situation, Dao and De, and that the field in which things are situated is itself made up of a variety of perspectives. From this point of view, the field itself is not objectified or universalized but has its own profound particularity, which is called Xuan De ([Ames and Hall 2003](), p. 157). When discussing the differences between Chinese and Western thought, he mentions that Western thinking and language are more transcendental, and Chinese culture is more characterized by associative thinking and associative language ([Ames 2002](), pp. 1–22). It can be said that this view is also reflected in Ames' understanding and translation of the concepts of Dao and De. In his understanding, Xuan De is a characteristic that embodies universal relevance, which implies the manifestation of Dao in different fields and perspectives.

Hans G. Moeller translates Xuan De as dark efficacy, and he mentions that "it is obvious that Dao and De are not creators in the strict sense of the word, but are rather, like the root of a plant, the 'force' within the cosmos that sustains all there is" and that "the cosmos is conceived of in terms of biological reproduction and fertility; it is understood as an 'organic' process of life. Dao and De are integral elements within that process and not an external origin" (Moeller 2007, p. 120). This is also the case with his understanding of Xuan De, which he sees as an affirmation of the inner power of growth that lies within all things themselves. Edmund Ryden's understanding is quite similar to Moeller's, as he translates Xuan De as an abstruse life force (Ryden 2008, p. 107).

The highly philosophical and insightful readings of Ames, Moeller, and Ryden have skillfully captured and presented the complex and profound connotations of De and Xuan De in the thought of the *Daodejing*. However, as stated at the beginning of this section, beyond the highly philosophical discussion, the proposal of Xuan De has a specific historical and theoretical background; that is, the Daoists attempted to construct a new ideology on ethics, self-cultivation, and political governance out of the tradition of bright virtue (*Ming De* 明德) that existed in the Western Zhou Dynasty and was inherited by Confucianism. In addition to Xuan De, the *Daodejing* contains similar concepts such as Shang De 上德 (the highest virtue), Guang De 廣德 (the broad-minded virtue), and Jian De 建德 (the most steadfast virtue), which are also used to highlight the differences and tensions between Daoism and the past ritual traditions, as well as Confucianism, and to reflect a new direction in Daoism's search for the question of De. If this is the case, it is obviously more appropriate to translate the De of Xuan De as virtue and Xuan De as dark virtue, just as D. C. Lau does (Lau 1963, p. 58), since such a translation is more capable of highlighting the different views of Confucianism and Daoism on the same concept of virtue.

If the above explanation is somewhat ambiguous in discussing and determining the translation of Xuan De, then we can discuss it further by looking for evidence and support in the *Daodejing* Chapter 38. An important concept mentioned in Chapter 38 is Shang De 上德. In the *Daodejing*, the meaning of Shang De is close to that of Xuan De; both refer to De that is different from ordinary ethics and morality. Chapter 38 contains the following passage:

> The highest virtue is not virtuous; therefore, it has efficacy.
>
> The lowest virtue does not let go of virtue; therefore, it has no efficacy.
>
> The highest virtue does not act and has no purpose.
>
> The highest humanity acts and has no purpose.
>
> The highest righteousness acts and has a purpose.
>
> The highest ritual propriety cats and nothing resonates with it, so that the sleeves are rolled up and coercion is exerted.
>
> Thus,
>
> After the Dao is lost, there is virtue.
>
> After virtue is lost, there is humanity.
>
> After humanity is lost, there is righteousness.
>
> After righteousness is lost, there is ritual propriety.
>
> (Moeller 2007, p. 93. Slightly modified.)
>
> 上德不德，是以有德；
>
> 下德不失德，是以無德。
>
> 上德無為而無以為；
>
> 下德為之而有以為。
>
> 上仁為之而無以為；
>
> 上義為之而有以為。
>
> 上禮為之而莫之應，則攘臂而扔之。

故失道而後德，失德而後仁，失仁而後義，失義而後禮。

Obviously, the highest virtue here is compared to the lowest virtue or even the virtues under the lowest virtue, such as benevolence, righteousness, and propriety. In this case, the meaning of De as potency, efficacy, nature, or power is far less obvious than that of a special kind of virtue. Therefore, the translation of Shang De as highest virtue is more appropriate than highest potency or highest efficacy. Based on the similarity of meaning between Xuan De and Shang De, the same idea applies when translating Xuan De.

## 5. Conclusions

In conclusion, De in the *Daodejing* is a very complex concept that not only carries traces of the evolution and development of Chinese philosophy but also highlights the dialogue and tension between Confucianism and Daoism. At the same time, it also contains rich philosophical implications concerning the theoretical connection between the concepts of De and Dao and thinghood. Because of the complex threads of thought contained therein, the translation of De is bound to be complicated and confusing. In addition to a contextualized and specific approach to the notion of De and to the book of *Daodejing*, some kind of coherent and synoptic understanding is also necessary. Just as the Daoists chose to reconstruct and restate the idea of De, which has a long history, perhaps the translation of De as virtue in the *Daodejing* can serve the same purpose. In other words, although the *Daodejing* also speaks of virtue, it is very different from the Confucian texts that speak of virtue in the sense of ethics and morality, and the different ways in which the same term is used precisely highlight the uniqueness and novelty of Daoist thinking. Of course, other interpretations and translations of the word De are also necessary and enlightening, as they also pinpoint the philosophical meaning of De at one level or another, enriching and deepening the reader's understanding of the concept of De.

**Funding:** This research was funded by "the Fundamental Research Funds for the Central Universities".

**Institutional Review Board Statement:** Not applicable.

**Informed Consent Statement:** Not applicable.

**Data Availability Statement:** Data are contained within the article.

**Conflicts of Interest:** The author declares no conflict of interest.

## Notes

[1]  As Misha Tadd pointed out in his project, Global Laozegetics (*Quanqiu Laoxue* 全球老學), many scholars inside and outside of China view translations of the Chinese classics as lacking fidelity to the original; however, translation is a realm of commentary and interpretation and "is not simply a flawed effort at reproducing a pristine text in a target language but a manifestation of the translator's inevitable interpretation of said text." See (Tadd 2022b).

[2]  Even if the translation of virtue in Confucian texts is relatively uniform, there are still many scholars who have paid attention to and discussed the various meanings of De in early Confucian texts. For example, see (Chan 2011).

[3]  It is worth noting, however, that even when translated as virtue, the difference between the meaning of De and virtue in Christianity is enormous. In the latter, virtue is primarily concerned with the moral or sexual dimension.

[4]  Nevertheless, there are scholars who are opposed to this idea, for example, see (Kryukov 1995, pp. 314–33).

[5]  For example, Sun Yirang 孫怡讓 identifies the graph as the initial graph of De and argues that the meaning of it should be virtue. See (Yu 1999, p. 2250).

[6]  For example, Yu Xingwu 于省吾 thinks that the graph should be *xun* 巡, see (Yu 1999, pp. 2251–56); Wen Yiduo 聞一多 argues that the graph is *sheng* 省, based on the rite of patrolling (*xun shou* 巡守) recorded in the *Book of Documents*. See (Wen 1993, p. 507).

[7]  About the detailed discussions on the historical event when King Wu conquered Shang, see, for example (Pines 2014; Dong 2021).

[8]  Sarah Allan suggests that De in this sense should be translated as favor or grace, which means the grace that is passed down in the form of heredity by Heaven or the lord on high. See (Allan 1997, p. 104).

[9]  See https://www.collinsdictionary.com/zh/dictionary/english/virtue (accessed on 1 January 2023).

[10] Due to the limitations of space, this article will not introduce the overall situation of the English translation of the *Daodejing*, but only select the more representative contents related to the theme of the article for discussion. For an overall discussion and analysis of the English translation of the *Daodejing*, see (Bebell and Fera 2000, pp. 133–47; Tadd 2022a, pp. 7–122).

[11] See, for example, De in chapters 21, 38, 41, 79. However, even if the term virtue is used in the translation of these chapters, its specific content is different from the ethics and morality emphasized by Confucianism.

[12] For the comparison between the connotations of virtue and Aretē, see (Zheng 2020).

[13] The English translation is by Ivanhoe and Norden. See (Ivanhoe and Norden 2000, p. 14).

[14] The translation is from Edmund Ryden and Brook Ziporyn and revised by the author. See (Ryden 2008, p. 107; Ziporyn 2023, p. 62).

[15] Lau, Chan, and Ivanhoe's translations of the first line of chapter 51 are "the Way gives them life, virtue rears them…Therefore, the myriad creatures all revere the Way and honor virtue", "Tao produces them (the ten thousand things). Virtue fosters them…Therefore, the ten thousand things esteem Tao and honor virtue", and "The Way produces them, virtue rears them…This is why the myriad creatures all revere the Way and honor Virtue", respectively. See (Lau 1963, p. 58; Chan 1969, p. 163; Ivanhoe and Norden 2000, p. 183).

[16] The translation is by Waston and slightly modified by the author.

[17] The translation is by the author. For the Chinese version, see (Lou 1980, p. 37).

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
