# Peer review of "The Understanding and Translation of De 德 in the English Translation of the Daodejing 道德經"

_religions, doi:10.3390/rel14111418_

Round 1
Reviewer 1 Report
Comments and Suggestions for Authors
This paper explores the English translation of the concept of "德" in the text of Laozi. It is evident that this translation issue is fundamentally related to the understanding of the concept of "德" in the context of Chinese philosophy, as well as the grasp of the meaning of terms such as virtue, potency, and inner power used to translate this concept in the context of Western philosophy. The study of this topic requires a high level of academic proficiency, and the author of this paper has done a commendable job in analyzing and organizing this issue.
The author fully recognizes the complex meanings accumulated by the concept of "德" in the development of ancient Chinese philosophy, and makes a certain sorting out of it, which provide necessary preparations for exploring the translation issue.
In the discussion on the translation of the concept of "德" in English, the author thoroughly compares different translations and analyzes the rationale behind each translation, taking into account the complex meanings of "德" in Chinese philosophical texts. Furthermore, the author examines the translation of "玄德" and points out that using the word "virtue" can better reflect the conceptual reconstruction of this complex concept in Daoism.
Additionally, the paper demonstrates the author's comprehensive and meticulous understanding of the relevant research in both the Chinese and English contexts under this topic.
There is also a suggestion here: Although the author primarily considers the unique meaning of "德" in its association with "道" and "物" when discussing the translation of "德" in the book of Laozi, the ultimate reason for choosing "dark virtue" as the translation for "玄德" is due to the concept of "德" having a special meaning of “virtue” in the discussion of "上德" and "下德" in Laozi. Could you provide some additional explanations for the special connotation of virtue that "德" holds in Laozi?
Author Response
Dear Reviewer,
Thank you so much for your comments and questions,they have been very enlightening for me.
Some further explanations and examples are added to exam the special connotation of "de" in the Laozi.
Reviewer 2 Report
Comments and Suggestions for Authors
This is a well-structured and clearly articulated article for the understanding of De. The author engages widely and deeply with Western and Chinese scholarship while providing his/her/their own critical comments.
I am only interested in seeing more under what contexts certain translations and interpretations win over others. For example, have people been interested in De because of their growing interest in ethics?
Author Response
Dear Reviewer,
Thank you so much for your precious feedback and for your question which is very interesting and important. The translation of "De" as "virtue" is absolutely relevant to the growing interest in in ethics :)
Reviewer 3 Report
Comments and Suggestions for Authors
It would be interesting to have a short note about the different meanings of De in Chinese context and the Western-Christian understanding which is mostly connected to moral/sexual virtue
Author Response
Dear Reviewer,
Thank you so much for the suggestion. And I totally agree that it is very interesting and important to compare "virtue" in the context of Chinese philosophy and Christianity. I added a footnote on it and will try to go into more details on this question!
Reviewer 4 Report
Comments and Suggestions for Authors
The research presented in this article spans multiple areas central to scholarship in Daoist philosophy:
--etymological roots of key philosophical terms (De, Xuan De)
--the cultural and political shift from the Shang to Zhou dynasties
--critical issues related to translation within different historical contexts
--comparisons of a range of English translations of the Daodejing text
--cross-cultural discussions of philosophies (Daoist, Confucian and Aristotelian)
--extensive scholarly references
Especially impressive is the author's demonstration of the dynamic nature Daoist and Confucian philosophies, which avoids the "Fixed Mind" tendency to view philosophy as a search for stagnant "eternal Truths."
Author Response
Dear Reviewer,
Thank you so much for your comments!